# Pathways by Which Self-Compassion Improves Positive Body Image: A Qualitative Analysis

**DOI:** 10.3390/bs13110939

**Published:** 2023-11-16

**Authors:** Trisha L. Raque, Brooke Lamphere, Christine Motzny, Julia Kauffmann, Kathryn Ziemer, Shaakira Haywood

**Affiliations:** 1Department of Counseling Psychology, University of Denver, Denver, CO 80210, USA; brookelamphere@gmail.com (B.L.); chrissy.motzny@du.edu (C.M.); juliakauffmannphd@gmail.com (J.K.); drshaakira@gmail.com (S.H.); 2Old Town Psychology, 1221 King Street, Alexandria, VA 22314, USA; drziemer@oldtownpsychology.com

**Keywords:** self-compassion, expressive writing, body image, college women

## Abstract

The current study applied consensual qualitative research–modified to essays written by 51 college women completing an expressive writing intervention over three time points for a total of 153 essays to identify how increases in self-compassion improve body image. A qualitative coding team tracked changes in affect and cognition over three time points. The results demonstrated that college women consistently expressed body acceptance and psychological flexibility. Additionally, the participants expressed important increases in mindfulness as well as decreases in social influences, feelings of separation, negative health behaviors, and attention to media messages. Decreases were found in their expressions of body functionality, love and kindness toward their body, and internal locus of control. These findings suggest pathways through which self-compassion may improve women’s body image by increasing mindfulness and decreasing the negative ways of relating to one’s body, specifically in the areas of media, clothing, make-up, and negative social interactions.

## 1. Introduction

Body image concerns are risk factors for developing eating disorders and are highly prevalent in college women [1,2,3], who more commonly report negative body image than men [4,5]. Furthermore, women are 10 times more likely to be diagnosed with an eating disorder compared to men [6]. Self-compassion, or the ability to be kind to oneself, experience a sense of common humanity, and practice mindfulness while facing suffering, has been found to protect against body image distress and to help cultivate a positive body image in college women [7]. Self-compassion may help women practice an attitude of acceptance toward their body and recognition that they are not alone in feeling that their body has flaws, rather than shame and isolation for not meeting our society’s unattainable standards of beauty [8,9]. Interventions aimed at increasing self-compassion have been identified as one way to enhance women’s body image [10,11,12]; yet, there is a need to identify the particular pathways through which such interventions work. Expressive writing is an easily accessible intervention format for evaluating changes in individuals’ affect and cognitions across time [13], and has a large body of evidence supporting its utility in enhancing physical and emotional well-being (e.g., [14]), including improving body image [15]. The current study applied consensual qualitative research–modified [16] to essays from a self-compassion expressive writing intervention to track changes in affect and cognition that may serve as mechanisms underlying self-compassion’s role in improving body image in college women. 

### 1.1. Body Image in College Women

Concerns around body image are disturbingly prevalent in college women, with 91% reporting that they have dieted to lose weight [17], and more than 50% of college women reporting that their body shape or weight influenced their self-worth to a moderate or extreme extent [18]. The most common onset of eating disorders overlaps with the traditional college years [19]. Thus, late adolescent and early adulthood have been identified as the optimal time for intervention to improve body image compared to other age groups [20]. Yet, Eisenberg and colleagues [21] found that 80% of college women with eating disorder symptoms fail to receive care. To reach a larger number of college women with body image concerns who may not otherwise have access to or seek treatment, online interventions have been proposed [19]. Such online interventions for reducing body image distress may pull from the body of research demonstrating their effectiveness with a wide range of female populations, including college women [22], breast cancer survivors [23], gynecological cancer survivors [24], and women with rheumatoid arthritis [25]. Moreover, given that up to 91% of Internet users first seek health-related information online [26], online interventions for addressing body image are needed. Earnhardt and colleagues [27] (p. 32) reported that for their online writing intervention targeting negative body image, their “recruitment waves have produced an overwhelming number of responses from campus women” who had body image distress but were not seeking treatment. They concluded that there is a need for further attention to online outreach to address college women’s body image concerns. Online body image interventions may allow college women to cultivate self-compassion in a “private and supportive environment” [22] (p. 259). This age group in particular has been identified as under-utilizing treatment to address body image distress [28], and an online format warrants exploration given the accessibility and anonymity of online interventions. In summary, the prior literature points to the prevalence of body image issues for college women and the need for additional efficient, easily accessible, private, and cost-effective online interventions, such as those incorporating self-compassion, to reach them.

### 1.2. Self-Compassion and Body Image

Self-compassion is an attitude comprised of self-kindness, mindfulness (i.e., staying present with one’s experiences in a non-judgmental manner), and common humanity (i.e., recognizing that one is not alone) in the face of suffering [29]. Struggles with body image may represent an ongoing experience of shame, self-criticism, and suffering in which one’s sense of self is threatened [12], similar to the experiences of suffering in which self-compassion is proposed to be most useful. Self-compassion has been identified as a promising protective factor against negative body image, with correlational studies finding links between self-compassion and fewer body concerns and eating-related guilt [30], disordered eating behaviors [31], internalization of the “thin ideal” [32], body shame, and body surveillance [33]. Furthermore, self-compassion was connected to lower levels of body objectification, body surveillance and anxiety about physique for women athletes [34], and lower levels of body dissatisfaction for breast cancer survivors [35]. These studies indicate the relevance of self-compassion for how women feel about their bodies.

Additional research has begun to conceptualize self-compassion as a moderator, or buffer, between body image distress and negative outcomes. Homan and Tykla [10] found that self-compassion moderated associations between body-related social comparisons and basing one’s self-worth on appearance with body appreciation. Breines, Toole, Tu, and Chen [8] found that those who approached a perceived body flaw with self-compassion reported lower levels of body shame, weight-gain concern, and self-punishment; and that body shame mediated the association between self-compassion and disordered eating. Self-compassion interventions directed at body image have amassed a growing body of evidentiary support. For example, Albertson, Neff, and Dill-Shackleford’s [36] three-week online self-compassion meditation intervention resulted in reduced body dissatisfaction, body shame, and body-related self-worth for the self-compassion group compared to a wait-list control group. Thus, self-compassion may be an effective point of intervention to create shifts in women’s negative body image.

In addition, self-compassion has been associated with the promotion of positive body image, which has been acknowledged as an important area worthy of study in its own right and uniquely predictive of well-being [37]. Body appreciation, the most comprehensive and examined aspect of positive body image [10], is defined as holding favorable opinions of one’s body despite perceived imperfections, awareness and attention to one’s bodily needs, engagement in health promotion behaviors, and rejection of the media’s portrayal of unrealistic bodily ideals [37,38]. Self-compassion has been connected to higher levels of intuitive eating [39,40], as well as body appreciation [10,30,36,41]. Furthermore, self-compassion in women has been associated with adherence to a broad conceptualization of beauty incorporating various body shapes and sizes [42]. Toole and Craighead [12] found that participants in a brief self-compassion online training reported greater changes in increased body appreciation, and decreased appearance-contingent self-worth and body surveillance compared to controls. Yet, their exploratory analysis suggested the changes in participants’ self-criticism, rather than in their self-compassion, were related to changes in the body image variables, except for body appreciation; decreasing self-criticisms was not enough to increase body appreciation. A six-week study on the effectiveness of an internet app focused on self-compassion reported increases in appearance esteem and self-compassion compared to a control group [43]. This growing body of evidence suggests that young women’s level of self-compassion may be critical to target to increase positive feelings for their bodies, and online interventions may be effective in doing so. 

In summary, self-compassion holds promise for protecting against and reducing negative body image and disordered eating [8], as well as promoting positive ways of relating to one’s body [22]. Researchers have suggested that self-compassion may help women decrease the emphasis they place on their appearance for their self-worth, decrease self-judgment for not meeting impossible standards of beauty, filter out social comparisons, remind themselves that everyone faces societal pressures connected to the thin-ideal (i.e., lower levels of isolation), embrace imperfections as part of being human, and increase respect toward one’s body [8,10,12]. Moreover, Tylka and Homan [44] suggest that women high in self-compassion may place greater importance on healthy behaviors and taking care of their bodies than on adhering to societal standards of beauty. However, these explanations for the mechanisms of self-compassion are based on researchers’ interpretations of quantitative findings, rather than explanations provided by women experiencing self-compassion. Additional research is needed to determine whether self-compassion interventions work by providing participants with a new way of responding to their body’s imperfections rather than specifically reducing body dissatisfaction [12]. Experimental and longitudinal designs are likewise needed to test hypotheses about how self-compassion may promote positive body image. Expressive writing interventions present an accessible and viable option of enhancing self-compassion and exploring the relationship between self-compassion and positive body image. Qualitative analysis of information gathered through expressive writing interventions may add richness to understanding of mechanisms of change facilitating the self-compassion and body image relationship.

### 1.3. Expressive Writing and Body Image

Expressive writing was developed by Pennebaker [45,46] and involved writing about a traumatic experience multiple times on a regular basis. It has been adapted to prompt writing about many different types of challenging issues, not all of which are single-event traumas; this adaptation has included expressive writing about one’s body, e.g., [47]. Expressive writing instructions generally direct participants to write about their deepest thoughts and feelings about an event, whereas instructions in control conditions direct participants to write about a neutral stimulus. For those experiencing negative body image, expressive writing may foster the restructuring and reorganization of their negative thoughts, thereby desensitizing participants to the thoughts and feelings that may fuel their negative way of relating to their bodies. 

Despite some indications of the benefits of expressive writing interventions, expressive writing interventions aimed at body image have generally reported mixed results. Arigo and Smyth [12] found that at an 8-week follow-up, college women in an expressive writing condition reported less sleep difficulty and less body-focused upward social comparison (i.e., comparing oneself to women thought to have “better” bodies) compared to the control condition, but no differences between the two groups were found in body image quality of life, disordered eating symptoms, general tendency for body comparisons, and body-focused downward comparisons (i.e., comparing oneself to women thought to have “worse” bodies). Others have found effects for both the expressive writing and control conditions. Frayne and Wade [48] found that both expressive writing and their control conditions reported a decrease in distress and only the control group reported a decrease in disordered eating. Earnhardt and colleagues [27] found that participants in both expressive writing and control conditions reported improved body image and mood, and decreased eating disorder behavior. Earnhardt and colleagues also noted that the expressive writing and control conditions may have “worked” in different ways. The writing condition may have served to offer desensitization and cognitive restructuring of negative thoughts about one’s body. In contrast, the control condition may have served as a form of distraction that decreased rumination on negative body image or increased their ability to assess objects more “objectively”, which they then may have applied to how they viewed their body. Researchers have urged for further evaluation of expressive writing, including the identification of effective variations in the writing paradigm, e.g., [48]. Self-compassion expressive writing represents one promising variation. 

### 1.4. Expressive Writing, Self-Compassion, and Online Interventions

As a result of the plethora of positive associations between self-compassion and mental and physical health (e.g., name withheld to preserve integrity of review process, 2011) and between expressive writing and mental and physical health, e.g., [14], researchers have begun to explore the potential of self-compassion expressive writing interventions. Self-compassion expressive writing interventions instruct participants to express self-kindness (e.g., express the same type of compassion toward yourself that you might express toward someone about whom you care), common humanity (e.g., keep in mind that all of us experience suffering as part of the human condition), and mindfulness (e.g., try to approach your experience with curiosity and acceptance). In experimental studies aimed at inducing self-compassion through expressive writing, the self-compassion writing groups reported significant reductions in depression and shame-proneness at 2-week follow-up compared to the traditional expressive writing and control group [49], and significant reductions in negative affect compared to the typical expressive writing group [50]. Baum and Rude [51]) found that depression-prone individuals in the self-compassion expressive writing group reported fewer depressive symptoms than those in the control condition. After inducing negative mood, Odou and Brinker [52] found that self-compassion writing improved mood more than the traditional expressive writing intervention. Thus, self-compassion may serve as an adaptive emotion regulation strategy when faced with difficulty [29]. However, Wong and Mak [53] failed to find significant group differences between self-compassion writing and control writing conditions in depression, self-compassion, or emotion regulation, although the self-compassion group alone reported significant decreases in physical symptoms at the 1- and 3-month follow-ups. They concluded that because self-compassion writing could serve as a cost-effective self-help intervention for promoting health, additional research is needed to explore the mechanisms that underlie this approach; our study heeds this call for additional research. 

There has been a growing utilization of online screening, prevention, and intervention tools to increase college student women’s engagement in innovative programs targeting body image issues, with promising results [19], including increased self-compassion [43]. Lipson and colleagues [19] suggest that online formats may improve treatment engagement by allowing participants to self-pace, access resources on their own time schedule (e.g., weekends), and improve their body image without a formal eating disorder diagnosis. Others note that online formats for self-compassion interventions allow participants to share sensitive body image worries without fear of judgment by others [54] and in a “private and supportive environment” [22] (p. 259). These online self-compassion interventions may build upon a long history of online expressive writing interventions, e.g., [47].

### 1.5. Aims of the Present Study

Building upon research on the potential utility of self-compassion for protecting against negative body image and increasing positive body image, the present study explored the mechanisms underlying a self-compassion expressive writing intervention. This is part of a larger study in which women from two universities were randomly assigned to a self-compassion expressive writing group (*N* = 51), a traditional expressive writing group (*N* = 50), or a control group that was prompted to reflect on the events of their day (*N* = 51) [55]. Each group followed a 20-min expressive writing prompt once per week for three weeks and completed pre- and post-test measures. Those in the self-compassion group demonstrated greater gains in their change score of self-compassion with a medium effect size compared to the other two groups, and quantitative changes in self-compassion predicted higher scores on measures of body appreciation, body image quality of life, and positive affect after controlling for baseline scores. To more fully identify the processes accounting for increased self-compassion in relation to body image, the current study qualitatively analyzed the essays completed by the 51 women in the self-compassion writing group. Consistent with qualitative methodologies that allow themes and ideas to emerge directly from the data rather than from the researchers’ preconceived understanding of the topic [56], no a priori hypotheses were generated in the current study. Instead, our objective was to qualitatively examine the pathways by which self-compassion improves body image in a multi-point essay delivery method of an online intervention.

## 2. Materials and Methods

### 2.1. Participants

Data for the current study were collected as part of a larger body image expressive writing study (names deleted to maintain integrity of review process, 2019). To more fully identify the processes accounting for increased self-compassion, the current study analyzed 153 essays completed over three time points written by the 51 women assigned to the self-compassion writing group (age M = 19 years, SD = 1.46; ethnicity, White = 84%, Asian = 4%, African American = 6%, Hispanic/Latinx = 4%, Unreported = 2%). Participants were all enrolled as undergraduate psychology students at a mid-Atlantic university (*N* = 21) and southern university (*N* = 27; 3 participants did not indicate which school they attended), and 45% were in their first year of college. Sixty-five percent of participants endorsed previous experiences with informal expressive writing (e.g., journaling, blogging), and 59% had written within the last six months. Participants were asked whether they had ever received counseling or mental health treatment for body image or disordered eating concerns (90% = no, 10% = yes).

### 2.2. Instruments and Qualitative Essays

The quantitative measures utilized in this expressive writing study are described in [55]. 

Participants were asked to complete three self-compassion writing exercises over one-week intervals for three consecutive weeks. Essays were completed online through Qualtrics, and participants were advised to write in a quiet and private area. Each exercise consisted of writing for 20 min in response to a prompt asking participants to write about their body image from a self-compassionate perspective as if they were expressing the kindness and understanding for themselves that they might receive from a good friend. They were also asked to consider what would help them not to feel alone in their experiences of body image and to accept any corresponding emotions. This prompt was based on previous research that explored self-compassion writing [57] and remained the same for all exercises.

### 2.3. Procedures

Participants were recruited from the undergraduate psychology student pool via a faculty collaborator at each university. They received USD 3 for completing the baseline survey, USD 3 for each of three writing exercises, and USD 5 for completing the final survey, for a possible total of USD 20. Compensation was provided in the form of gift cards, and participants also received research credit when applicable. Participants gained access to the study either by locating the study link on their institution’s research page or by receiving an email from a member of the research team. After initial contact, participants were asked to complete informed consent documentation via Qualtrics Online Survey software before completing inclusion criteria and baseline measures. Informed consent was obtained from all subjects involved in the study. Inclusion criteria for the study included identification as an undergraduate college woman over the age of 18, and not currently receiving counseling or treatment for concerns related to body image or disordered eating. Institutional Review Board approval was obtained from all involved institutions prior to participant recruitment (Protocol 769061-3). Informed consent was obtained from all subjects involved in the study. To protect participants’ confidentiality, all data are protected and held by the primary researcher according to the retention guidelines of the American Psychological Association.

Participants were randomly assigned to one of three expressive writing conditions and only essays from the participants assigned to the self-compassion writing condition were included in the current study. Once a week for three consecutive weeks, participants received a link via email to complete their writing exercise on Qualtrics. Participants were sent two reminder emails if they did not complete their essays within 24 and 48 h of receiving the link. If participants did not complete a writing exercise within the one-week time period allotted for each, they were excluded from further participation in the study. All 51 included participants completed all surveys and writing exercises. 

### 2.4. Data Analysis

#### 2.4.1. Researchers

A team of four coders in pairs of two analyzed the data. All coders were women affiliated with a counseling psychology doctoral program (one associate professor, four doctoral students) and had clinical experience working with body image issues and mindfulness. Three coders and the external auditor identify as White, European Americans; one coder identifies as White, European American, and Native American, and one researcher who helped with management of coding responses but did not serve as a coder identifies as Black. No coders had personally experienced eating disorders, and no coders were instructors for study participants. 

#### 2.4.2. Qualitative Analysis (CQR-M)

Data were analyzed utilizing consensual qualitative research—modified [CQR-M; 16]. CQR-M is a version of Hill et al.’s [58] original CQR method modified to be more compatible with large amounts of written data. Initially, the coding team worked together to code essays from two randomly selected participants, discussing potential themes (termed “domains” in CQR-M) and categories until consensus was achieved. The domains and categories were derived directly from the participants’ words, and each line of data was analyzed. After creating a coding summary sheet that represented the consensus version of domains and categories, each coding pair was randomly assigned half of the essays. Each pair then engaged in the following coding process: (1) each coder independently read each essay, utilizing annotations to make note of key observations or themes at the level of each line; (2) each coder completed a coding summary sheet to identify domains and categories for each essay; (3) each coding pair met to compare and discuss independent codes; (4) each coding pair came to a consensus on domains and categories for each essay, completing a final coding summary sheet. The consensus process is a central component of all CQR methodologies, as the inclusion of multiple perspectives minimizes potential biases in interpretation and is helpful in capturing the complexity of qualitative data [16,58]. Coding meetings involved continuous revisions to the original coding list, with revisions then discussed by the entire team for approval. 

The research team also included an independent auditor who provided feedback on the coding process at regular intervals. This auditor also reviewed the finalized codebook, providing feedback such as suggestions for merging categories. All feedback from the auditor was reviewed and integrated by the research team. Once the finalized codebook was approved, domain and category frequencies and percentages were calculated across cases for each of the three time points [16]. Each line from an essay could receive only one code. If the domain or category was represented in a full line within the essay, then that essay was counted as containing the domain or category; thus, essays could be counted in multiple domains and categories. 

Spangler et al. [16] recommended utilizing a difference of 30% as the criteria for determining an “important difference” between time points; in this case, an important change over the course of the intervention. Research has documented the prevalence of increased emotional distress immediately following participation in expressive writing exercises [48], along with greater potential for positive influence of expressive writing over time [59]. Consequently, we calculated the percent change between essays at time point one (T1) and time point 3 (T3) with the following formula: [(Time 3% − Time 1%)/Time 1%) × 100]. 

## 3. Results

A total of seven domains were identified, with a description of and illustrative quotes for each provided in Table 1. 

We calculated percentage change between T1 and T3 for all domains and categories to identify areas of participant experience that changed over the course of the intervention. A complete list of domains, categories, percentage of total essays containing each domain, frequencies across time points, and percent change in frequencies over time is included in Table 2. Each of the domains will be covered in the sections below in the order by which they were most commonly coded. Direct participant quotations are presented for findings that were represented in over 50% of total essays or that fit criteria as an “important finding”. All participants are represented by a pseudonym to protect their confidentiality.

### 3.1. Positive Body Image

When asked to write about their body image from a self-compassionate perspective, positive body image was consistently the most common theme across all three writing exercises. A positive body image involved appreciation and acceptance of one’s body as opposed to criticism or shame. Within this domain, a high percentage of participants wrote in a way that reflected acceptance of their body. For example, one participant wrote, “I have come to accept my body, flaws and all” (Zoe), while another noted, “I can only accept the cards that I have been dealt in life and learn to adjust or love them…Nobody is perfect and that makes us unique” (Mary). 

While acceptance remained high throughout the course of the writing, the following two categories within the domain of positive body image were identified as important changes: focus on body functionality and love and respect. Participant focus on body functionality decreased by 64% from T1 to T3. The category focused on body functionality included expressions of gratitude or pride for physical characteristics or abilities of one’s body, such as athleticism, strength, exercise, or movement. For example, Deana expressed gratitude for everything her body allows her to do, remembering that many people do not share her experience: “I am strong and I should be grateful that I could do anything that I want to. I can climb mountains, snowboard, rock climb, run, and more. Some people struggle with simple everyday tasks. I need to appreciate my strength and my beauty”. Others expressed gratitude for their body’s ability to keep them healthy and to recover from injury or illness. 

The category of love and respect included expressions of appreciation for one’s body in the form of affirmations or positive self-talk. The percentage of participants expressing love and respect for their bodies decreased by 36% from T1 to T3. Many participants made clear statements about body parts that they like, admire, or appreciate, emphasizing the importance of focusing on positive aspects of the self instead of the negative. Others discussed the importance of accepting imperfections in the body as the path to loving oneself: “I just think that imperfections are your own perfections. It’s what makes you different and unique. I love myself” (Margaret).

### 3.2. Psychological Flexibility

Psychological flexibility also remained consistently high across the three writing sessions, and on average across the three time points, was represented in 71% of the essays. Psychological flexibility involved an expression of connection to the present moment, and awareness of one’s ability to change or alter thinking, decisions, and behavior related to the body. Perspective-taking was represented in 50% of all essays and included thoughts about minimizing the importance of body image in context (e.g., “there are other more important things to think about”); and participants’ ability to defuse from, or positively reframe, negative body image thoughts: “Weight tells us nothing. It doesn’t tell me whether I’m healthy or not. It says nothing about who I am as a person. It’s a number. I do not need to be defined by a number. I cannot be summed up in a number. I am worth so much more than that” (Beckie).

The category of mindfulness increased by 41% from T1 to T3, indicating an important change. Expressions of mindfulness included non-judgmental awareness of body image, noticing how one thinks about one’s body, and reminding oneself to practice self-compassion toward one’s body. For example, June explained, “I need to remember that it’s okay to feel unhappy sometimes, but it’s very important to let those feelings pass. Recognize that they occur, and move forward. It’s easy to get stuck in a pattern of negative self-image, but it’s not necessary”. 

### 3.3. Empowerment

Sixty-six percent of essays noted feelings of empowerment over the course of self-compassion writing. Empowerment involved finding strength or confidence in oneself, especially in taking control of one’s life. Internal locus of control was identified as an important finding as this category decreased by 40% between T1 and T3. Discovering an internal locus of control entailed cognitively recognizing what is within one’s power to change and what is not: 

“I think that once you love yourself the way you are it is a lot easier to take control of the situation and change the things that you may not like about yourself”(Jenna).

### 3.4. Social Context

This domain included external social factors that influenced participants’ body image, and incorporated categories with a positive social influence, with a negative social influence, and the impact of the media or socially portrayed ideals of beauty on participant’s body image. This domain was mentioned in 61% of all essays and decreased by 39% between T1 and T3. The category of *negative social influences* involved experiences with social relationships (e.g., friends, family) that participants perceived as critical or judgmental, and decreased by 62%. For example, Ksenia described how she started to pay attention to the way that other people impact her view of herself, choosing not to give others’ opinions so much power: 

“I know I get really self-aware when I think that someone gave me a funny look and I automatically think it’s because I look funny or something must be wrong…In the past when I surrounded myself with negative people or found out that people I thought I was friends with weren’t sincere it really affected the way I felt about myself but as I have gotten older I have learned how to handle this and how to make sure people I include in my life are actually there for me.”

Others described the role of a critical parent or important other in negatively influencing their feelings about their body: “It would help if my mother didn’t make me feel worse for not having less acne or for not having the perfect teeth. I know she is trying to help but when she points out my flaws it just makes me feel inadequate to everyone else around me” (Rose). As a result of social influences, participants described feeling unworthy of connection, inadequate, or ignored due to their physical appearance. Others spoke of experiences with stigmatizing social expectations related to gender identity and body image. For example, Clara noted that, “Women are more often than not looked at for their physical appearance more than what they have to say, or what’s in our brains. It’s sad but it is the way our culture is”. 

The influence of the *media* on participants’ body image included descriptions of constant inundation from media messages portraying seemingly flawless, and often culturally-bound, expectations of beauty as a source of stress that was harmful to their body image. This category decreased 31% from T1 to T3. Beckie explained: “I…believe that in today’s society, the media as well as other influential people in one’s life (celebrities, models, parents, peers, etcetera) set standards that in many cases are unattainable. I believe that is a major reason as to why many people, especially adolescents, have negative self-images”. Others noted how much they appreciated companies with advertising campaigns that included many body types, not just those considered ideal, championing hope for change and a more compassionate, accepting social context.

### 3.5. Negative Body Image

The negative body image domain captured cognitive, social, and affective themes that had a negative impact on participants’ view of their body image. Sixty-two percent of participants wrote about aspects of negative body image during the study, and important changes occurred in the categories of feeling separate from others and negative health behaviors. These categories decreased by 60 and 100 percent, respectively, by T3. Feeling separate from others involved feelings of isolation due to body image. For example, some participants felt isolated because they looked different than others. Kim spoke about feeling different from her peers because of her body type, and her desire to fit in: “I look around and see all these girls who look the same and as much as I kinda enjoy standing out, it gets tough. I want to be able to fit in you know? Even if it’s just for a second for one little thing”. Others spoke of their unwillingness to conform to social expectations such as dieting or rigorous exercise and feeling different as a result. Still, others felt isolated due to physical characteristics such as height or hair color that they perceived as abnormal. 

While reported in only 5% of essays, some participants endorsed negative health behaviors that contributed to negative body image, with this category decreasing over time. These described behaviors included disordered eating behaviors (e.g., bingeing, restricting, purging), intense dieting, or excessive exercise. This was described by Mary as, “Whenever I am feeling like I am fat (which usually happens after I eat bad food or a big meal) I want to rush to the gym”.

### 3.6. Material Influences

The domain Material Influences, represented in 23% of all essays, included mention of the influence of clothing and make-up on participants’ feelings about themselves. Participants expressed how they relied on these external materials to improve self-confidence and body image: “It feels so right when I put on that perfect dress or pair of pants and I feel good-looking or dare I say hot. At that moment, I feel like nothing can get me down as I put on just the right amount of makeup” (Mary). The mention of the use of materials to bolster body image decreased by 46% from T1 to T3, and the categories of make-up and clothing decreased by 43 and 50% respectively. For example, Deana captured this as, “If I buy makeup and skincare products and actually try and pick out an outfit instead of throwing on a t-shirt, I can look pretty”.

### 3.7. Disengagement

The final domain encompassed disengagement from the writing topic or prompt. Feelings of disengagement were expressed through statements of boredom (e.g., “I am bored of this topic”), the avoidance of discussion of self-compassion or body image (i.e., choosing to talk about something else that either depersonalized the prompt or was unrelated), and saturation (i.e., specific note by participants that they had nothing left to say on the topic). The disengagement domain increased by 300% from T1 to T3; however, it was only coded in 18% of total essays.

## 4. Discussion

In this qualitative study of essays written by college women in a self-compassion expressive writing intervention, the results demonstrated that participants consistently expressed body acceptance and psychological flexibility in the form of perspective-taking across time. Additionally, they expressed important decreases in negative social influences, feelings of separation, negative health behaviors, external bases for body image (i.e., make-up, clothing), and attention to media messages. They also exhibited decreases in social comparisons and body shame that approached the cutoff for designation as an “important” change [16]. Such findings are consistent with those reported by Seekis et al. [22]) regarding their online mindful self-compassion intervention that resulted in improved body image, including significant reductions in social appearance anxiety and upward appearance comparisons. The present findings also replicate those of Turk and colleagues [9] on the importance of self-compassion interventions for reducing body shame. 

Further, in the current study, unexpected decreases were reported in their focus on body functionality, expressions of love and kindness toward their body, and internal locus of control. Collectively, these findings suggest pathways through which self-compassion may improve body image; specifically, self-compassion may increase women’s ability to approach their body in a mindful and accepting manner that decreases feelings of isolation and emphasis given to negative social and societal messages. 

Self-compassion expressive writing interventions appear to serve as an accessible, online method for improving the way that young women relate to their bodies. Thus, our findings indicate that this type of intervention may overcome some of the barriers to accessing treatment to improve body image reported by college women [19]. Our participants were not required to have a formal eating disorder diagnosis and could complete their writings on their own schedules in their preferred private space, which expanded beyond who may traditionally be able to access body image interventions [22]. Study retention and engagement were high, indicating our intervention was accessible. Further, unexpected findings in categories such as body functionality and expressions of love and kindness toward one’s body suggest that participants could present their feelings about their bodies authentically in the online writing format, without feeling pressured to conform to social desirability around maintaining a façade of body positivity Increased accessibility and promotion of authenticity appear to be clear benefits of our online writing intervention format. 

Participant essays consistently conveyed high levels of body acceptance and perspective-taking from the first writing prompt through the third prompt. In the context of this study, body acceptance represented a nonjudgmental attitude toward one’s body, including acceptance of any perceived imperfections. Body acceptance was distinct from conceptualizations of body appreciation, e.g., [10] in that acceptance did not always imply favorable feelings. For example, in contrast to the findings of Toole and Craighead [12] on the role of self-criticism as the mechanism of change in decreasing negative body image, our study found that body self-criticism remained consistent at T1 and T3, whereas body acceptance increased. Participants in the present study increased their awareness of their feelings toward their body and accepted those feelings for what they were, even when they were self-critical, rather than trying to transform them into positive feelings. Interventions aimed at body acceptance have been identified as key evidence-based practices for improving the body image of people with eating disorders [60] and for decreasing body dissatisfaction and negative affect [61], and self-compassion appears to be an important aspect of prompting body acceptance. 

A similar form of acceptance, combined with a lack of rumination, was captured by high levels of perspective-taking. The participants acknowledged how they felt about their bodies without becoming overly attached to those feelings, instead recognizing that their relationship with their bodies was only one aspect of their life experience. The consistently high presence of these two nonjudgmental aspects of relating to one’s body did not suggest that a self-compassionate approach to body image was about movement toward positivity. A self-compassionate approach involved acknowledgment without rumination on one’s current feelings and attitude toward one’s body, even when they were negative, and placing them in a larger context such that their body image did not define the way they experience their identities in their entirety. 

The significant increase in mindfulness expressed in essays from the first to the third prompt further illustrates how self-compassion may be embodied in a nonjudgmental awareness of one’s feelings, thoughts, and experience of their body. As participants practiced self-compassion and their mindfulness increased, they became increasingly conscious of how they felt about their body but did not become consumed by those feelings. This phenomenon, which we referred to as mindfulness within the context of body image, resembles the construct of body image flexibility, defined as “the capacity to openly and freely experience body dissatisfaction and other relevant disordered eating thoughts without making efforts to avoid or change them” [62] (p. 665). Body image flexibility has been associated negatively with body dissatisfaction [63] and disordered eating behaviors [62,63,64,65]. Within the context of expressive writing, body image flexibility or mindfulness related to body image thoughts are in line with cognitive processing theory (CPT; [66]). CPT outlines how the writing experience allows for the creation of a reorganized, adaptive, and coherent narrative that has integrated initial affective and cognitive reactions to the writing stimulus [66]). Body image related mindfulness or body image flexibility may serve as the reorganizing mechanism that provides the cognitive space to integrate one’s experiences, feelings, and thoughts about one’s body without becoming overly attached to them; with mindfulness representing a beneficial point of intervention for body image concerns [67]. In short, a more flexible and mindful acceptance of their current body image appeared to be the resulting narrative from the self-compassion expressive writing intervention. 

The reorganized narrative resulting from increased body-image mindfulness or body image flexibility may help explain the unexpected decreases in their focus on body functionality and an internal locus of control for improving their body image. Participants also reported decreases approaching importance in their feelings of ownership, or their intention to take the necessary actions to change the aspects of their body that they did not like. The categories of body functionality, internal locus of control, and ownership imply a conditional acceptance of one’s body based on the body’s ability to serve a function (e.g., be strong) or change (e.g., with enough exercise, the body will improve). Perhaps decreases in these areas indicate a more balanced form of body acceptance of what the body is, rather than what it does or could be. Internal locus of control and ownership were coded within the “empowerment” category. Often increasing empowerment is viewed as a positive change, yet our findings capture the nuance within how empowerment may be enacted. For example, empowerment may go awry in situations in which individuals feel empowered to hold an internal locus of control that blames themselves for things that they do not like about their bodies or they feel empowered to engage in disordered eating (e.g., restricting behaviors). Our findings indicate the importance of exploring how college women are enacting empowerment and the extent to which it promotes body acceptance or appreciation. 

However, these findings are in contrast with the small percentage of participants who reported a decreased engagement in negative health behaviors over the course of the intervention. Tylka and Homan [44] theorized that self-compassion may encourage individuals to take improved care of themselves. A small meta-analysis conducted by Sirois, Kitner, and Hirsch [68] found that self-compassion positively related to health-promoting behaviors, with a high positive affect and a low negative affect serving as separate mediating pathways. Within the context of body image, perhaps the unconditional acceptance of one’s body allows individuals to have cognitive and affective resources more readily available to direct toward improving their health. However, further experimental and longitudinal research is needed to examine empirically how increased self-compassion may influence motivation and behavioral change directed at improving body image. 

Unexpected decreases also were found in participants’ expressions of love and kindness toward their bodies. This category captured expressions of affirmation toward one’s body. Perhaps strivings for a positive body image may have influenced what participants felt pulled to write in their initial essays, as they were aware that they were participating in a body image study, and they were given directions asking them to treat themselves with kindness. By the final writing exercise, participants may have been more aware of and willing to disclose how they currently feel about their bodies rather than what they aspired for their relationship with their body to be. Perhaps the online format in their private space allowed them to convey their feelings about their bodies in an authentic manner. 

While participants decreased their focus on social, societal, and external influences (i.e., clothing, make-up) on their body image, they also decreased in their feelings of separation from others. Additionally, decreases in comparisons (−28%) and shame (−22%) approached the criteria for designation as an important change. These findings suggest that self-compassion could impact how participants view themselves in relation to others, and the extent to which their interactions with others affect the way they felt about themselves. As participants became more aware of the ways in which they related to their body and became more connected to their own cognitions and affect, they also became increasingly aware of their social environment, including both positive and negative social influences. The decreased reference to these social influences may indicate that participants took steps to minimize them. Simultaneously, their feelings of shame and embarrassment about their bodies and the extent to which they compared themselves to others decreased, as did the extent to which they felt their body image prevented them from feeling connected to others. Such findings are consistent with prior research on associations between higher self-compassion and decreased body shame [8,36,69,70] and body surveillance [33,34] as pathways through which self-compassion may affect body image. 

Furthermore, social integration theory [71] proposes that expressive writing changes how participants relate to and interact with others, typically increasing their need to express the insights gained by expressive writing to others. For instance, Frattaroli’s [72] meta-analysis found that participation in expressive writing resulted in increased social disclosures about one’s trauma, thereby creating greater opportunity for increased social support and social integration. In the current study, the ways in which participants’ social interactions were affected included decreasing the emphasis they placed on how others felt about them and their social comparisons, thereby minimizing their experiences of shame and decreasing their feelings of isolation due to their body image. Rather than increasing positive social interactions, the experience of a self-compassion expressive writing intervention may result in participants distancing or buffering themselves from social interactions that affected their body image negatively. These results are similar to correlational studies finding that self-compassion moderated associations between body comparison and body appreciation; self-compassion may help women filter out cultural messaging about the desirability of the thin ideal such that they could practice a more accepting and kind way of relating to their body [10].

In their systematic review of 28 studies on self-compassion and negative body image, Braun, Park and Gorin [7] identified the following four pathways through which self-compassion may operate: (a) direct impact; (b) preventing the onset of risk factors; (c) buffering the effects of risk factors, and (d) halting the mediational path through which risk factors may operate. In summary, the present study suggests that self-compassion may have a direct impact on body acceptance, perspective-taking, and body-image-related mindfulness or body image flexibility. Self-compassion appears to have a buffering effect on the negative influence of social and societal messaging about standards of beauty, and approached a nearly important halting of the mediational path of shame and social comparison. 

Regarding methodology, the current study’s high participant retention rate and the depth of reflection embodied in the essays highlight the utility of a self-compassion expressive writing intervention as an easily accessible format. Given the changes across the three time points and the lower level of many categories reported at time point 2, the use of three points for data collection appears important for longitudinal analysis of expressive writing. Moreover, the current study suggests the utility of applying CQR-M to analyze expressive writing essays as a means of identifying mechanisms of change. Concepts related to authentic ways of relating to self and decreased importance placed on social influences would have been more difficult to identify had another methodology been employed without incorporating a qualitative coding process. 

### Limitations

Although the use of a self-compassion writing prompt and CQR-M represent a novel research design, their limitations should be acknowledged. The representativeness of the findings based on the current sample warrants additional evaluation, as there may have been a participant self-selection bias for those who found writing helpful before the intervention or a skewness toward those psychologically-minded based on the recruitment process. Participants were largely White women in psychology classes, potentially contributing to demand characteristics, limiting the transferability of findings, and indicating the need for future research with more diverse samples. Further, as participants were excluded if they were currently in treatment for an eating disorder and disordered eating symptomology was not assessed in the current sample, it is unclear as to what level of body image concerns are captured in the current sample. 

The expressive writing structure includes specific limitations. It allows participants to write about whatever they identify as relevant. As represented in the 29 identified categories in this study, the elicitation of such a wide range of responses may pose challenges to coding and identifying higher-order themes that fully represent the data. Further, 14% of our participants appeared to have saturated the topic by their third essay, with 26% exhibiting a form of disengagement by the end of the writing exercise. Although such disengagement is expected over the course of a longitudinal study and may reflect the difficulty of the emotionally charged content, it also points to the need to consider variations in the self-compassion expressive writing format (e.g., the time between the writing prompts). 

The designation of the 30% cutoff was pre-determined by Spangler et al. [16] but may be arbitrary for some research purposes. Moreover, we examined change using percentage scores for all participants rather than tracking change at the level of the individual participant. Although the three time points allowed us to identify changes over time, these were composite changes within the group of 51 participants rather than for individual participants. Additional longitudinal research at the individual level may offer alternative explanatory mechanisms for self-compassion expressive writing.

## 5. Conclusions

This study uniquely contributes to the existing literature by highlighting the pathways through which self-compassion may operate to improve body image. Women instructed to write about their bodies in a self-compassionate way were consistently able to channel positive body image, body acceptance, empowerment, and find a balanced perspective. In addition, the way that they related to their body changed over the course of the writing exercises, suggesting that self-compassion may improve body image by increasing mindfulness and decreasing negative ways of relating to one’s body, specifically in the areas of media, clothing, make-up, and negative social interactions. These findings hold important implications about the power of self-compassion to not only buffer negative influences but also improve young women’s relationships with their bodies. Additionally, the findings contribute to the growing body of research supporting the utility of online self-compassion interventions for expanding treatment approaches for college women with body image concerns. 

## Figures and Tables

**Table 1 behavsci-13-00939-t001:** Domains and categories from CQR-M analyses of self-compassion essays.

Domain	Category	Description	Illustrative Quotes
Positive Body Image		Positive aspects of relationship with one’s body or perspective of body	
	Acceptance	Acceptance of body as is, with imperfections	As of currently, I’m not really broken. I’m a little overweight; I’ve got some unhealthy eating habits. I need improving. But I’m not broken.
	Functionality	Focus on what the body or body parts allow one to do	I want to have a body that will be useful.
	Love and Respect	Appreciation for one’s body	You are only given one body in this life so we should love it and treat it with respect.
	Connection with Others	Feeling close or desiring to help others as a result of feeling good about one’s body	I can still be there for young girls who aren’t so comfortable with themselves and tell them that they are beautiful the way they are.
	Self-worth	Positive body image has a reciprocal relationship with self-esteem	I feel better about myself as a person when I feel good about my body.
Psychological Flexibility		Self-awareness of one’s feelings about the body, and willingness to change or alter thinking, decisions, or behavior	
	Adaptability	Comfortable with changes in and feeling about one’s physical body	I know to look at my body image as an ongoing thing.
	Mindfulness	Non-judgmental awareness of how one thinks about or relates to one’s body	I don’t really know how to be compassionate towards my body other than to say that it’s okay and not too bad.
	Perspective-Taking	Minimizing the importance of body image in context	Life is way too short trying to be someone else and to look like someone else.
	Mind-Body Connection	Awareness of connection between mind and body	I work out because it relieves stress and makes me feel better about myself.
	Common Humanity	Recognition of shared human experiences with body image	I know I’m not the only one that thinks this way; in fact, odds are good that the majority of people have this state of mind.
	Spirituality	Influence of religious or spiritual beliefs on one’s relationship with body	God tells me that I’m his beloved child. That I was fearfully and wonderfully made. That I have an eternal hope in him. I can find my identity in the truth that I am his creation, that he delights in me…
Empowerment		Feelings of increased confidence, authority, or power regarding body image	
	Internal Locus of Control	Cognitively recognizing what is within one’s power to change	I think that once you love yourself the way you are it is a lot easier to take control of the situation and change the things that you may not like about yourself.
	Ownership	Taking action to change aspects of body that one does not like	I have been going to the gym more frequently, putting in 100 percent effort at workouts, and eating less or not so late at night or least try to. It is a hard struggle I face but I hope the outcome is worth it in the end. I just want to feel comfortable in my body like I did two years ago.
	Fake It until You Make It	The belief that if one works to have positive body image, then one will eventually have it	I have to repeatedly tell myself you are not ugly, and hopefully one day I will believe it.
	Advice	Giving advice to others about how to cope with or improve body image	Love your body, love yourself, love others, love life.
Social Context		Factors in social environment that influence body image	
	Positive Social Influence	Relationships with important others involve elements of positive body image	My boyfriend makes me feel like his most prized possession here on earth. He forever fills me with words of love, reminding me of my beauty, the way I am loved for exactly who I am, and how he wants to spend his forever alongside mine.
	Negative Social Influence	Relationships have a negative influence on body image	I think her [friend’s] eating behavior rubbed off onto me, and I started viewing myself as fat and ugly.
	Media Influence	Portrayals of body image in media impact body image	Who has defined what a person is “supposed” to look like anyways? The media? The answer is yes. We get so caught up in photo-shopped images of people who make their money by being beautiful and wonder why we don’t look exactly like them.
Negative Body Image		Cognitive, social, and affective components of negative body image	
	Feeling Separate from Others	Feelings of isolation due to body image	I know I have a larger butt and wide hips so I tend to feel different than anyone else…
	Self-Criticism	Negative self-talk about body image	I am always criticizing it [my body], putting the wrong foods in and being lazy.
	Comparisons	Comparing one’s body to others’	Comparing my body to others at field hockey is something that I’ve always done.
	Shame	Feeling shame or unworthiness related to body image	I think my biggest fear is that because of how I look nobody is going to love me fully.
	Negative Health Behaviors	Endorsing extreme or disordered eating behaviors	I’ve done extreme diets in the past, I’ve stopped eating, even thrown up after guilty meals.
Material Influence		Material influences on body image that are derived from sources that are not part of the body it self	
	Appearance: Makeup	Makeup utilized to change or alter appearance	The right makeup makes me feel good.
	Appearance: Clothing	Clothes or outfits influence how one feels about the body; clothes are related to both body image and self-worth	When I dress up and get ready I know I look good. I feel good and I look good.
Disengagement		Specific mention of disengagement from intervention, prompt, or topic	
	Boredom	Participant explicitly expressed disinterest in writing	I am bored.
	Avoidance	Participant avoided discussion of topics included in prompt	…anyway I digress.
	Saturation	Participant felt that they have shared everything they can about the topic of body image	I have nothing else to say about this.

**Table 2 behavsci-13-00939-t002:** Percentage of essays that endorsed domains and categories from the CQR-M analysis of self-compassion expressive writing essays by time point.

Domain	Category	% of Total Essays	Time 1 (% Essays)	Time 2 (% Essays)	Time 3 (% Essays)	% Change (T1–T3)
Positive Body Image		78	86	73	77	−11
	Acceptance	58	65	51	57	−12
	Functionality	24	33	26	12	−64
	Love and Respect	50	61	49	39	−36
	Connection with Others	14	16	16	12	−25
	Self-Worth	21	20	26	18	−10
Psychological Flexibility		71	77	69	69	−10
	Adaptability	18	16	20	20	25
	Mindfulness	22	22	14	31	41
	Perspective-Taking	50	53	45	51	−4
	Mind-Body Connection	17	26	20	24	−8
	Common Humanity	43	43	39	45	5
	Spirituality	7	8	6	8	0
Empowerment		66	71	71	57	−20
	Internal Locus of Control	29	33	33	20	-40
	Ownership	39	41	43	31	−24
	Fake It ’til you Make It	3	4	2	4	0
	Advice	35	29	39	35	17
Social Context		61	77	61	47	−39
	Positive Social Influence	34	35	41	26	−26
	Negative Social Influence	19	26	24	10	−62
	Media	22	26	24	18	−31
Negative Body Image		62	67	57	63	−6
	Feeling Separate from Others	7	10	6	4	−60
	Self-Criticism	41	47	33	43	−9
	Comparisons	35	43	31	31	−28
	Shame	17	18	20	14	−22
	Negative Health Behaviors/ Illness	5	8	6	0	−100
Material Influence		23	26	29	14	−46
	Appearance: Makeup	14	14	22	8	−43
	Appearance: Clothing	12	16	12	8	−50
Disengagement		18	6	22	26	333
	Boredom	1	2	2	0	−100
	Avoidance	7	4	10	8	100
	Saturation	7	2	6	14	600

## Data Availability

Data that are not presented in the article are unavailable due to privacy restrictions.

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
