# Peer review of "Pathways by Which Self-Compassion Improves Positive Body Image: A Qualitative Analysis"

_behavsci, 2023, doi:10.3390/bs13110939_

Round 1

Reviewer 1 Report

Comments and Suggestions for Authors

Manuscript ID: behavsci-2671930

Title: Pathways by which Self-Compassion Improves Positive Body Image: A Qualitative Analysis

The manuscript needs some modifications so that it could be better than before. It would be helpful if the authors would consider the following points:

Major points:

- The paper suffers from a poor English structure throughout and cannot be published or reviewed properly in the current format.

- Please pay attention to the definite and Indefinite Articles "The, a, an" in the text

- Please polish the abstract

- Update the references as there is no reference from 2020 to 2023

- The hypothesis of the study should be clarified at the end of the Introduction section.

- The novelty of the study needs to be highlighted compare to other similar studies.

- Put the objective of the study at the end of the introduction section.

- Make a Figure to show the research design

- The discussion section needs to be rewritten and updated for the references used.

- Insert the correct format style for journals in the references in the text and references list.

Minor points:

- Line 11: Change " affect" to " effect"

- Line 12: Change " three time" to " three-time"

- Line 16: Add "that" after " These findings suggest"

- Line 49: Change "has" to "have"

- Line 52: Change "health related information" to "health-related information"

- Line 56: Add "an" before "under-utilizing treatment"

- Line 121: Add "with" after "participants"

- Line 153: Add "the" before "identification"

- Line 176: Delete "a" before " self-compassion" and before " control writing"

- Line 232: Change "via by " to "via"

- In Table: Change "I need improving." To "I need to improve"

- Review the grammar accurately in the contents of Table 1 because it has many grammatical problems.

- Line 305: what do you mean? " We calculated percent change"

- Line 326: Change " importance" to " important"

- Line 328: Change " focus" to " focused"

- Lines 337,383: Change " The category" to " The category of"

- Line 339: Change " their body" to " their bodies"

Comments on the Quality of English Language

Extensive editing of English language required

Author Response

Reviewer 1:

Major points:

- The paper suffers from a poor English structure throughout and cannot be published or reviewed properly in the current format.

Response:  Thank you for this feedback.  We have revised throughout, attending to your points.

- Please pay attention to the definite and Indefinite Articles "The, a, an" in the text

Response:  We have revised accordingly.

- Please polish the abstract.

Response:  We have revised the abstract for improved clarity. 

- Update the references as there is no reference from 2020 to 2023

Response:  Thank you for that suggestion.  We have now incorporated the following citations to more fully capture recent research:

  1. Alleva, J. M., Diedrichs, P. C., Halliwell, E., Peters, M. L., Dures, E., Stuijfzand, B. G., & Rumsey, N. (2018). More than my RA: A randomized trial investigating body image improvement among women with rheumatoid arthritis using a functionality-focused intervention program. Journal of Consulting and Clinical Psychology, 86(8), 666–676. https://doi-org.du.idm.oclc.org/10.1037/ccp0000317.supp (Supplemental)
  2. Barbeau, K., Guertin, C., Boileau, K., & Pelletier, L. (2022). The effects of self-compassion and self-esteem writing interventions on women’s valuation of weight management goals, body appreciation, and eating behaviors. Psychology of Women Quarterly, 46(1), 82–98. https://doi-org.du.idm.oclc.org/10.1177/03616843211013465
  3. Byrne, M. E., Eichen, D. M., Fitzsimmons-Craft, E. E., Barr Taylor, C., & Denise E. Wilfley, D. E. (2016). Perfectionism, emotion dysregulation, and affective disturbance in relation to clinical impairment in college-age women at high risk for or with eating disorders, Eating Behaviors, 23, 131-136,  https://doi.org/10.1016/j.eatbeh.2016.09.004.
  4. Eisenberg,D.,Nicklett,E.,Roeder,K.,&Kirz,N.(2011).Eating disorder symptoms among college students: Prevalence, persistence, correlates, and treatment-seeking. Journal of American College Health, 59(8), 700–707.
  5. Ghaderi, A., Welch, E., Zha, C., & Holmes, E. A. (2022). Imagery rescripting for reducing body image dissatisfaction: A randomized controlled trial. Cognitive Therapy and Research, 46(4), 721–734. https://doi-org.du.idm.oclc.org/10.1007/s10608-022-10295-z
  6. Goel, N. J., Sadeh-Sharvit, S., Trockel, M., Flatt, R. E., Fitzsimmons-Craft, E. E., Balantekin, K. N., Monterubio, G. E., Firebaugh, M.-L., Wilfley, D. E., & Taylor, C. B. (2021). Depression and anxiety mediate the relationship between insomnia and eating disorders in college women. Journal of American College Health, 69(8), 976–981. https://doi-org.du.idm.oclc.org/10.1080/07448481.2019.1710152
  7. Lipson, S. K., Jones, J. M., Taylor, C. B., Wilfley, D. E., Eichen, D. M., Fitzsimmons-Craft, E. E., & Eisenberg, D. (2017). Understanding and promoting treatment-seeking for eating disorders and body image concerns on college campuses through online screening, prevention and intervention. Eating Behaviors, 25, 68–73. https://doi-org.du.idm.oclc.org/10.1016/j.eatbeh.2016.03.020
  8. Messer, M. Lee, S., & Linardon, J. (2023). Longitudinal association between self-compassion and intuitive eating: Testing emotion regulation and body image flexibility as mediating variables.  Journal of Clinical Psychology, 79, 2625-2634. Doi:  1002/jclp.23659
  9. Mifsud, A., Pehlivan, M. J., Fam, P., O’Grady, M., van Steensel, A., Elder, E., Gilchrist, J., & Sherman, K. A. (2021). Feasibility and pilot study of a brief self-compassion intervention addressing body image distress in breast cancer survivors. Health Psychology and Behavioral Medicine, 9(1), 498–526. https://doi-org.du.idm.oclc.org/10.1080/21642850.2021.1929236
  10. Rodgers, R. F., Donovan, E., Cousineau, T., Yates, K., Mcgowan, K., & Cook, E., et al. (2018). BodiMojo: Efficacy of a mobile-based intervention in improving body image and self-Compassion among adolescents. Journal of Youth and Adolescence, 47(7), 1363–1372. http://dx.doi.org/10.1007/s10964-017-0804-3
  11. Seekis, V., Bradley, G. L., & Duffy, A. L. (2020). Does a Facebook-enhanced Mindful Self-Compassion intervention improve body image? An evaluation study.  Body Image, 34, 359-269.  Doi:  1016/j.body/im.20200.07.006
  12. Trachtenberg, L., Wong, J., Rennie, H., McLeod, D., Leung, Y., Warner, E., & Esplen, M. J. (2020). Feasibility and acceptability of i‐Restoring Body Image after Cancer (i-ReBIC): A pilot trial for female cancer survivors. Psycho-Oncology, 29(4), 639–646. https://doi-org.du.idm.oclc.org/10.1002/pon.5288
  13. Turk, F., Kellett, S., & Waller, G. (2021). Determining the potential link of self-compassion with eating pathology and body image among women: A longitudinal mediational study.  Eating and Weight Disorders—Studies on Anorexia, Bulimia, and Obesity, 26, 2683-2691.  Doi:  1007/s40519-021-011441-
  14. Turk, F., Kellett, S., & Waller, G. (2023). Testing a low-intensity single-session self-compassion intervention for state body shame in adult women: A dismantling randomized controlled trial.  Behavior Therapy, 54, 916-928.  doi:  1016/j.beth.2023.04.001.

- The hypothesis of the study should be clarified at the end of the Introduction section.

Response:  As we explain in the section “Aims of the Present Study,” we follow the protocol of qualitative research designs and do not have study hypotheses; instead, the results arise from the data.  To help clarify our study focus, we have added the sentence “Instead, we qualitatively examined what themes captured the pathways by which self-compassion improves body image.”

- The novelty of the study needs to be highlighted compare to other similar studies.

Response:   Thank you for this suggestion.  We have revised both the introduction and the discussion to more clearly highlight the unique contributions of our study.

- Put the objective of the study at the end of the introduction section.

Response:  We appreciate this suggestion. To address this, we have added the following sentence, “Instead, our objective was to qualitatively examine the pathways by which self-compassion improves body image.”

- Make a Figure to show the research design

Response:  We think that Table 1 better captures our research process than a figure could.  As a qualitative study, we are unclear as to what type of figure would be helpful. We are open to any suggestions for more specific guidance as to what type of figure.

- The discussion section needs to be rewritten and updated for the references used.

Response:   Thank you for this feedback.  We have updated the references throughout the manuscript, including in the discussion, and rewritten parts of the discussion to more fully capture how our findings relate to prior research and what our study uniquely adds.

- Insert the correct format style for journals in the references in the text and references list.

Response:  We were a bit unclear as to what was considered the correct style.  On this webpage for Behavioral Sciences (https://www.mdpi.com/journal/behavsci/instructions ), it states that “Your references may be in any style, provided that you use the consistent formatting throughout. It is essential to include author(s) name(s), journal or book title, article or chapter title (where required), year of publication, volume and issue (where appropriate) and pagination….When your manuscript reaches the revision stage, you will be requested to format the manuscript according to the journal guidelines.”   We currently have references listed using APA formatting.  Would it be preferred for us to switch to numbering citations in the order in which they appear, as is indicated in this website:  https://www.mdpi.com/authors/references?  We are happy to revise the references as needed.

Minor points:

- Line 11: Change " affect" to " effect"

Response:  We are referring to affect as emotions, so the correct word is “affect” in this line.

- Line 12: Change " three time" to " three-time"

Response:  Thank you for that feedback. We have made that revision.

- Line 16: Add "that" after " These findings suggest"

Response:  We have revised this sentence, adding “pathways through which” for improved clarity.

- Line 49: Change "has" to "have"

Response:  We have made that change.

- Line 52: Change "health related information" to "health-related information"

Response:  We have made that change.

- Line 56: Add "an" before "under-utilizing treatment"

Response:  Under-utilizing is a verb in this sentence and therefore we have not added an “an” before it.

- Line 121: Add "with" after "participants"

Response:  We have made that change.

- Line 153: Add "the" before "identification"

Response:  We have made that change.

- Line 176: Delete "a" before " self-compassion" and before " control writing"

Response:  We have made that change.

- Line 232: Change "via by " to "via"

Response:  We have made that change.

- In Table: Change "I need improving." To "I need to improve"

Response:  This was a direct quote from a participant, and thus we have not edited the participant’s direct statement.

- Review the grammar accurately in the contents of Table 1 because it has many grammatical problems.

Response:  As noted above, this Table includes direct quotes from participants’ writings.  Our intent is to capture participants’ quotes exactly as they expressed them, so we have not edited their quotes in any ways.

- Line 305: what do you mean? " We calculated percent change"

Response:  We explain on lines 300-303 how we calculated percentage change. Specifically, “we calculated the percent change between essays at time point one (T1) and time point 3 (T3) with the following formula: [(Time 3% - Time 1%) / Time 1%) x 100].”

- Line 326: Change " importance" to " important"

Response:  Thank you for catching that error. We have made that edit.

- Line 328: Change " focus" to " focused"

Response:  Focus in this line is a noun, not a verb, and thus we have not made this edit.

- Lines 337,383: Change " The category" to " The category of"

Response:  We have made that edit.

- Line 339: Change " their body" to " their bodies"

Response:  Thank you for catching that error. We have made that edit.

Reviewer 2 Report

Comments and Suggestions for Authors

107-124: You do a great job of not only showing the relevance of self-compassion to body image struggles, gathering good research, but also provide strong indication for why qualitative studies are needed. 

126-133: Pennebaker's expressive writing is still the standard, but still looks at trauma through a lens predominantly informed more by event based trauma rather than the more common forms of relational trauma explored (most recently) by Gabor Mate. Because body image may not be anchored in a single event, another sentence exploring how his work was adapted but not relied upon exclusively may be helpful.

171: Should be fewer, not less, I would imagine.

174-180: Grammar is a bit difficult to follow ("a...conditions"). Overall, 1c through 1e do a good job of extending your argument. You ground a clear methodology and the reason for suspending a hypothesis in a compelling framework.

259-265: I appreciate your identification of the researchers in terms of ways they identify. It provides a grounding initial vantage of how research is approached. 

PP 7-12: I like the example quote per category and the blend of qualitative and quantitative work. 

Section 3 felt like a thorough analysis of the data, offering both a wide view of numbers and depth through specific comments. 

L 483-486: Excellent distinction

L 524-533: I'd been wondering about this variable in the data and how it'd be interpreted--I think you offer compelling insight into why these decreases entail an important step forward. 

L 546-563: Similarly here--you're doing a good job of articulating questions about the data, especially given the lack of a hypothesis you'd be driven to confirm, and offering what seem like solid explanations. You've earned a lot of good will at this point, which makes it easy to respect both the challenges you find posed and your response to these challenges.

L 610-634: The discussion of limitations also shows that the authors are reflective, self-critical, and balanced in their estimation of what works well and what needs to be kept in mind. 

Overall, this seems thorough, well-done, and vital work. Thank you!

Author Response

Reviewer 2

107-124: You do a great job of not only showing the relevance of self-compassion to body image struggles, gathering good research, but also provide strong indication for why qualitative studies are needed. 

Response:  Thank you for that feedback.

126-133: Pennebaker's expressive writing is still the standard, but still looks at trauma through a lens predominantly informed more by event based trauma rather than the more common forms of relational trauma explored (most recently) by Gabor Mate. Because body image may not be anchored in a single event, another sentence exploring how his work was adapted but not relied upon exclusively may be helpful.

Response:  This is an important consideration when utilizing expressive writing.  To address this, we have revised the manuscript to read “Expressive writing was developed by Pennebaker (1989, 2007) and involved writing about a traumatic experience multiple times on a regular basis. It has been adapted to prompt writing about many different types of challenging issues, not all of which are single-event traumas; this adaptation has included expressive writing about one’s body (e.g., Barbeau et al., 2022). 

171: Should be fewer, not less, I would imagine.

Response:  Thank you for catching that error. We have made that edit.

174-180: Grammar is a bit difficult to follow ("a...conditions"). Overall, 1c through 1e do a good job of extending your argument. You ground a clear methodology and the reason for suspending a hypothesis in a compelling framework.

Response:  We appreciate this feedback.  We have edited this section in attempt to improve the grammar and clarity. 

259-265: I appreciate your identification of the researchers in terms of ways they identify. It provides a grounding initial vantage of how research is approached. 

Response:  Thank you for noting this.

PP 7-12: I like the example quote per category and the blend of qualitative and quantitative work. 

Response:  We appreciate this feedback.

Section 3 felt like a thorough analysis of the data, offering both a wide view of numbers and depth through specific comments. 

Response:  We are glad this section felt thorough.

L 483-486: Excellent distinction

Response:  Thank you for noting this.

L 524-533: I'd been wondering about this variable in the data and how it'd be interpreted--I think you offer compelling insight into why these decreases entail an important step forward. 

Response:  Thank you for this feedback.

L 546-563: Similarly here--you're doing a good job of articulating questions about the data, especially given the lack of a hypothesis you'd be driven to confirm, and offering what seem like solid explanations. You've earned a lot of good will at this point, which makes it easy to respect both the challenges you find posed and your response to these challenges.

Response:  We appreciate your comments on our communication of our results.

L 610-634: The discussion of limitations also shows that the authors are reflective, self-critical, and balanced in their estimation of what works well and what needs to be kept in mind. 

Response:  Thank you for noting this.

Overall, this seems thorough, well-done, and vital work. Thank you!

Reviewer 3 Report

Comments and Suggestions for Authors

Great start on a very interesting and needed area of topic. The aim of your study is to do a process evaluation of this intervention if I’m understanding correctly. The increase in mindfulness conveyed in the essay from the first to third writing exercise has the potential to show self-compassion is a mechanism promoting change in body image. Self-compassion could be accessed through forms of expressive writing and has the potential to increase psychological benefits within this realm for college-age women. Some possible areas of improvement within this article are as follows.

Abstract

      Clarify in abstract re number of essays vs cases and time pointes used in analysis in the abstract

Introduction

      If possible find more current citations relating to your topic more specifically related to body image concerns within the college population (Croll 2002, Striegel 2008, and Drewnowski 1995, Earnahrdt et al 2002). As well as citations related to internet use to seek health information (Bessel et al 2003).

      I would suggest discussing the role that a fairly homogenous sample of young, college, relatively white women in a psychology class might have had on your findings. And/or why this target population has increased risk/need.

      Psychology students might already have a more progressive understanding on how expressive writing could aid their self-compassion. Is this something that was considered when selecting this group?

      Many of your citations are a bit more summary statements of articles. Shift wording so I know why that citation/study is important to your argument that self-compassion and body image are connected. Make sure each paragraph has a concluding sentence to connect it to the next concept/paragraph. The goal of each section should be to provide a clear explanation of what we know, what we don’t know and what we are trying to find out!

      1a.

      This paragraph starts moves from scoping body image concerns for women to a need for online intervention. This seems too big a leap or is missing connections between sentences. Why is the internet a viable intervention, is it desired? Is it effective? Is it where they are at? Does your intervention need to take place online to be effective?

      Statement on “immense response to intervention” appears to be about rate of treatment for body image concerns. Need to clarify or reword.

        It might be helpful to have the procedures information explaining your intervention before the aim of the study since your data collection procedures are your intervention as well.

        Aim of study.

        N should consistently refer to number of essays or number of people as participants.

        More specifics on what “demonstrated greater gains in self-compassion” means from baseline to post would be helpful.

 Materials and methods

      Within this group 65% of students had partaken in informal expressive writing and 59% had written in the last 6 months. Do you think this had an effect on the results of the study compared to populations who do not partake in the form of study? Yes or no explain why this might have an impact.

      2b. Make the prompt more clear, perhaps put in quotations? Were these separate questions or asked together? Were they the same questions each time?

      2d. I appreciate the disclosure- it might be more helpful to know if any have experience with disordered eating as well as if any researchers were instructors of the students at the time of essay?

      I would like to know more about the coding process- were full lines coded, paragraphs or phrases? Could things get multiple codes or just one?—how do you calculate % of an essay, is this just if it was included or not at all in the essay or does frequency matter?

Data Analysis

  • Although this article is written for a psychology journal about the use of psychologists and there is one outside auditor coming from a nutritional expertise in one category of empowerment; ownership, Eating less in order to lose weight and hoping that the outcome is worth it in the end and trying to be someone she was two years ago, would not match empowerment in the eyes of a dietitian.  While the descriptions resonate with taking action, action can also mean feeling lack of power/confidence (disordered eating or othorexia), especially as the authors noted at this young age/stage. May just be the example included.

Discussion

      Consider wording generalizing conclusions to all young women. Your initial aim of understanding the mechanisms for change and gaining deeper perspective speak to potentially transferable knowledge, but not necessarily generalizable/representativeness as you would with quantitative data.

Author Response

Reviewer 3

Great start on a very interesting and needed area of topic. The aim of your study is to do a process evaluation of this intervention if I’m understanding correctly. The increase in mindfulness conveyed in the essay from the first to third writing exercise has the potential to show self-compassion is a mechanism promoting change in body image. Self-compassion could be accessed through forms of expressive writing and has the potential to increase psychological benefits within this realm for college-age women. Some possible areas of improvement within this article are as follows.

Abstract

  • Clarify in abstract re number of essays vs cases and time pointes used in analysis in the abstract

Response:  Thank you for this feedback.  We have added this information to the abstract.

Introduction

  • If possible find more current citations relating to your topic more specifically related to body image concerns within the college population (Croll 2002, Striegel 2008, and Drewnowski 1995, Earnahrdt et al 2002). As well as citations related to internet use to seek health information (Bessel et al 2003).

Response:  We agree that this is important.  We have now incorporated an additional 14 citations, many of which focus on the college population (e.g., Byrne et al., 2016; Eisenberg et al., 2011; Goel et al., 2021; Lipson et al., 2017) and on online health engagement (e.g., Seekis et al., 2020).

  • I would suggest discussing the role that a fairly homogenous sample of young, college, relatively white women in a psychology class might have had on your findings. And/or why this target population has increased risk/need.

Response:  Thank you for this point.  We have had updated citations in the introduction (Bryne et al., 2016; Gael et al., 2021) that documents the vulnerability of college women for eating disorders.  Within section 1a., we also added how the most common onset of eating disorders overlaps with the traditional college years (Lipson et al., 2017).  Moreover, we have elaborated in the limitations section to address by this by now reading, “Participants were largely White women in psychology classes, potentially contributing to demand characteristics and limiting the generalizability of findings and indicating the need for future research with more diverse samples.”

○      Psychology students might already have a more progressive understanding on how expressive writing could aid their self-compassion. Is this something that was considered when selecting this group?

Response:  We agree that this is an important consideration. In the limitation sections, we have included this as part of the potential bias stemming from the recruitment process.

  • Many of your citations are a bit more summary statements of articles. Shift wording so I know why that citation/study is important to your argument that self-compassion and body image are connected. Make sure each paragraph has a concluding sentence to connect it to the next concept/paragraph. The goal of each section should be to provide a clear explanation of what we know, what we don’t know and what we are trying to find out!

Response:  Thank you for this feedback.  We have revised accordingly, including adding connecting sentences between paragraphs. We think these edits clarify how past research holds relevance and establishes a rationale for what our study aims to uncover.  

  • 1a.

○      This paragraph starts moves from scoping body image concerns for women to a need for online intervention. This seems too big a leap or is missing connections between sentences. Why is the internet a viable intervention, is it desired? Is it effective? Is it where they are at? Does your intervention need to take place online to be effective?

Response:  We agree that this section warrants revision to make a stronger argument for why online interventions are needed.  We have revised this paragraph to indicate the large number of college women experiencing body image distress but not receiving treatment, and how online interventions may draw from a large body of efficacy research with other female populations.  The intervention does not need to take place online to be effective, but the online format may increase its reach to those who may not otherwise receive any form of treatment.

○      Statement on “immense response to intervention” appears to be about rate of treatment for body image concerns. Need to clarify or reword.

Response:  We have clarified in the manuscript to better capture the “overwhelming number of responses”  that Earnhardt and colleagues (2002, p. 32) experienced from non-treatment seeking college women for their online intervention for negative body image.  

  • It might be helpful to have the procedures information explaining your intervention before the aim of the study since your data collection procedures are your intervention as well.

Response:  We thank you for this feedback.  We have elaborated in the Aims section to hopefully clarify the intervention and the focus of the current study.

  • Aim of study.

○        N should consistently refer to number of essays or number of people as participants.

Response:  Thank you for drawing our attention to this. We have revised throughout to use N only to refer to number of people as participants.

○        More specifics on what “demonstrated greater gains in self-compassion” means from baseline to post would be helpful.

Response:  We have revised this to indicate that we are referencing that change scores in self-compassion were higher for the self-compassion group, and this was at a medium effect size. 

Materials and methods

  • Within this group 65% of students had partaken in informal expressive writing and 59% had written in the last 6 months. Do you think this had an effect on the results of the study compared to populations who do not partake in the form of study? Yes or no explain why this might have an impact.

Response:  We think this could indicate a self-selection bias for recruitment.  We have expanded upon this in the limitations section.

  • 2b. Make the prompt more clear, perhaps put in quotations? Were these separate questions or asked together? Were they the same questions each time?

Response:  The text currently summarizes the prompts rather than quotes them directly to reduce word count.  We are happy to include the full writing prompt in an appendix, if that would be deemed helpful. The prompt was the same each of the three writing sessions.  The instructions were “For the three writing sessions, we would like you to write about your body image from a self-compassionate perspective. Self-compassion means to be kind to yourself and to be less self-critical or self-blaming. Try to write in a way that expresses understanding, kindness, and concern to yourself the way you might express concern to a friend who has expressed similar feelings. Write about the many ways you can think of in which other people also experience similar feelings to the ones you describe. We would like you to write whatever comes to you, but make sure the writing provides you with what you need in order to feel understood and not alone in your experiences related to your body image. We realize that individuals may feel a wide range of emotions about their bodies, and we want you to write from the perspective of someone who is accepting of these emotions.

  • 2d. I appreciate the disclosure- it might be more helpful to know if any have experience with disordered eating as well as if any researchers were instructors of the students at the time of essay?

Response:  Thank you for this comment.  We have now added, “No coders had personally experienced eating disorders, and no coders were instructors for study participants.”

○      I would like to know more about the coding process- were full lines coded, paragraphs or phrases? Could things get multiple codes or just one?—how do you calculate % of an essay, is this just if it was included or not at all in the essay or does frequency matter?

 Response:  Full lines were coded into one category, but one essay (containing many lines) may capture multiple categories.  If the category was represented in a full line, then that was counted as that essay containing the category; the category did not need to be represented multiple times in order to count. We have elaborated on this in section 2d.

Data Analysis

  • Although this article is written for a psychology journal about the use of psychologists and there is one outside auditor coming from a nutritional expertise in one category of empowerment; ownership, Eating less in order to lose weight and hoping that the outcome is worth it in the end and trying to be someone she was two years ago, would not match empowerment in the eyes of a dietitian.  While the descriptions resonate with taking action, action can also mean feeling lack of power/confidence (disordered eating or othorexia), especially as the authors noted at this young age/stage. May just be the example included.

Response:   We appreciate these points.  To better convey empowerment, we have included a different participant quote in text.  More specifically, we have included this quote instead, ““I think that once you love yourself the way you are it is a lot easier to take control of the situation and change the things that you may not like about yourself” (Jenna).

Discussion

  • Consider wording generalizing conclusions to all young women. Your initial aim of understanding the mechanisms for change and gaining deeper perspective speak to potentially transferable knowledge, but not necessarily generalizable/representativeness as you would with quantitative data.

Response:  This is an important consideration given our qualitative research design.  We have revised the discussion with attention to not over-generalizing our findings (e.g., revising wording to read “may” and “could” to be more tentative).

Thank you for the opportunity to revise our paper.  We look forward to your feedback, and thank you for your time and consideration.

Round 2

Reviewer 3 Report

Comments and Suggestions for Authors

Thank you for your updates!

I would recommend a few additional minor updates before acceptance.

-I'm still uncertain if the online delivery method might not warrant it's own separate introduction section, perhaps after you explain the variable explanation of target audience to support why this delivery method is being used with this target audience on these topics.  In the least, I would include connection to multi-point online essay delivery method as a component of your study aims and what makes it unique/particularly applicable to study further.

-I also think there is benefit to including your full prompt in an appendix for future replicability!

-Methods: Could each line receive multiple codes or just one? How did % difference take into account different lengths or numbers of codes within an essay?

-Re: above about online format, this is a particular emphasis in your introduction, but relatively limited discussion in the implications and conclusion. Please be sure to reconnect to that topic as I think it also contributes to the uniqueness of your study findings and implications!

-I appreciate the change in quote in the Empowerment finding and I agree feeling the self-efficacy to take action is good! I would like to see a bit in the discussion section talking about how this could be a support or detriment depending on what actions they are empowered to follow (more disordered eating or less).

Author Response

Dear Editorial Board of Behavioral Science,

We thank you for the opportunity to submit a revised manuscript (Manuscript behavsci-2671930) “Pathways by which Self-Compassion Improves Positive Body Image: A Qualitative Analysis” to Behavioral Sciences.  We appreciate the thoughtful recommendations of Reviewer 3, and have responded below.

Reviewer 3 -I'm still uncertain if the online delivery method might not warrant it's own separate introduction section, perhaps after you explain the variable explanation of target audience to support why this delivery method is being used with this target audience on these topics.  In the least, I would include connection to multi-point online essay delivery method as a component of your study aims and what makes it unique/particularly applicable to study further.

Response:  Thank you for this feedback.  We have revised the last section before our stated aims to elaborate further on the significance of the online delivery method.  We have changed that section heading 1d to “Expressive Writing, Self-Compassion, and Online Interventions” and added a final paragraph on this topic.

Reviewer 3 -I also think there is benefit to including your full prompt in an appendix for future replicability!

Response:  We are happy to do this, and have uploaded the Appendix with the full writing prompt instructions with this revision.

Reviewer 3  -Methods: Could each line receive multiple codes or just one? How did % difference take into account different lengths or numbers of codes within an essay?

Response:  Each line could receive only one code, and we have added this information in text.  However, we did not account for different lengths or number of codes within an essay. 

Reviewer 3  -Re: above about online format, this is a particular emphasis in your introduction, but relatively limited discussion in the implications and conclusion. Please be sure to reconnect to that topic as I think it also contributes to the uniqueness of your study findings and implications!

Response:   Thank you for that feedback.  Within the third paragraph of the discussion and on the paragraph on expressions of love and kindness, we have expanded to note how the online intervention offered increased accessibility and promotion of authenticity as participants addressed body image concerns.  Additionally, we have added a final sentence to the conclusion that addresses our how findings contribute to the growing body of research supporting the utility of online self-compassion interventions for expanding treatment approaches for college women with body image concerns.  We hope this better conveys our study implications and the uniqueness of our findings.

Reviewer 3 -I appreciate the change in quote in the Empowerment finding and I agree feeling the self-efficacy to take action is good! I would like to see a bit in the discussion section talking about how this could be a support or detriment depending on what actions they are empowered to follow (more disordered eating or less).

Response:   We agree that this is an important point.  We have elaborated on it in the discussion section on page 12, hopefully better capturing the nuance of how empowerment is enacted. 

Thank you for the opportunity to revise our paper.  We look forward to your feedback, and thank you for your time and consideration.